# Genetic characterization of a new radish introgression line carrying the restorer gene for Ogura CMS in *Brassica napus*

Tonghua Wang[1,2], Yiming Guo[2,3], Zengxiang Wu[1], Shengqian Xia[1], Shuijin Hua[4], Jinxing Tu[1]*, Mei Li[2]*, Weijiang Chen[2]

1 National Key Laboratory of Crop Genetic Improvement, Huazhong Agricultural University, Wuhan, China, 2 Crop Research Institute, Hunan Academy of Agricultural Sciences, Changsha, China, 3 Rapeseed Engineering Research Center, Ministry of Education, Huazhong Agricultural University, Wuhan, China, 4 Institute of Crop and Nuclear Technology Utilization, Zhejiang Academy of Agricultural Sciences, Hangzhou, China

☯ These authors contributed equally to this work.
* tujx@mail.hzau.edu.cn (JT); limei1230@126.com (ML)

**Data Availability Statement:** The data discussed in this manuscript have been deposited to NCBI's SRA database with BioProject, number PRJNA488570 and is accessible through the

## Abstract

Creating a homologous restorer line for Ogura cytoplasmic male sterility (Ogu-CMS) in *Brassica napus* is meaningful for the wider application of Ogu-CMS system in rapeseed production. Previously, an independent development of a new Ogu-CMS restorer line (CLR650) was reported locally from crossing between *Raphanobrassica* (AACCRR, 2n = 56) and *B. napus* and a new version of Ogu CMS lines CLR6430 derived from CLR650 was characterized in this study. The results showed that the fertility restoration gene in CLR6430 presented a distorted segregation in different segregating populations. However, the majority of somatic cells from roots had a regular chromosome number (2n = 38) and no radish signal covered a whole chromosome was detected using GISH. Thirty-two specific markers derived from the introgressed radish fragments were developed based on the re-sequencing results. Unique radish insertions and differences between CLR6430 and R2000 were also identified through both radish-derived markers and PCR product sequences. Further investigations on the genetic behaviors, interactions between the fertility restoration and other traits and specific molecular markers to the introgression in CLR6430 were also conducted in this study. These results should provide the evidence of nucleotide differences between CLR6430 and R2000, and the specific markers will be helpful for breeding new Ogura restore lines in future.

## Introduction

Cytoplasmic male sterility (CMS) is a widespread inherited trait in plants, which is controlled by an incompatibility between the nucleus and mitochondrial genes and has been widely used for F1 seed production. Some CMS systems have been identified in *B. napus*, such as Polima CMS (*pol* CMS) [1], Ogura CMS (*ogu* CMS) [2], the common *B. napus* CMS (*nap* CMS) [3],

website linkage at http://www.ncbi.nlm.nih.gov/sra/
SRP159151.

**Funding:** Funding for operating costs of this
research was provided both by the National Natural
Science Foundation for Young Scientists of China
(Grant No. 31401467), The Key Research and
Development Program of Hunan Province (Grant
No. 2016NK2200) and The Fundamental Research
Funds for the Central Universities (Grant No.
2662020ZKPY019).

**Competing interests:** The authors have declared
that no competing interests exist.

and a novel type introgressed from Chinese woad (*inap* CMS) [4]. Ogu CMS was first discovered in Japanese radish (*Raphanus sativus*) from an unknown cultivar (Ogura 1968), and the sterility of o*gu* CMS was stable in various climate conditions [5], which used widely in hybrid breeding of important *Brassica* crop worldwide..

Creation of the corresponding restore lines in oilseed *Brassicas* could achieve three-line breeding system in rapeseed which is widely applied in the industry, however the whole process has been difficult as the restorer gene (*Rf*) only existed in some Europe and Japanese wild radish, but not in *B. napus* nor any *B. rapa* varieties. Therefore, the *Rf* gene had to be introduced from radish into oilseed rape through intergeneric hybridization. Homoeologous recombination between R and A or C genome were achieved through polyploidy cross [6], gamma irradiation of donor pollen [7], or asymmetric protoplast fusion [8]. However, along with the introduction of the *Rfo* gene into *B. napus*, extra segments of radish genome was introgressed into *B. napus* which brought undesirable traits for breeders, such as elevated glucosinolate level and poor seed set (the number of seed per silique) [9]. The early version of *Brassica* Ogu restorer lines developed by INRA identified a close linkage between glucosinolate content and the *Rfo* gene [10], suggesting more difficulties in breeding double low cultivars. Therefore, following research was dedicated for the improvement of the restorer line. Delourme [11] reported a low glucosinolate (9–18 μmol/g) Ogu-CMS restorer line in *B. napus* assisted by RAPD markers, however the *Rfo* loci was heterozygous and not heritable to siblings. Successful break-through of low glucosinolate Ogu-CMS restorer line was obtained through gamma radiation when crossing with *B. napus* of low glucosinolate content, and named as R2000 [7, 11]. Other independent breeding of *B. napus* Ogu restorer line using radiation or other methods were reported by Pioneer Hi-bred Company and Syngenta Biotechnology Company [12–14], with shortened radish fragment in new generations.

The *Rfo* gene, *orf687*, encoding a 687-amino-acid protein with multiple pentatricopeptide repeats, alters the expression of the sterile gene *orf138* at post-transcriptional level [15, 16]. The gene has been localized physically and mapped in the distal region on linkage group $N_{19}$ in Pioneer-derived Ogu-CMS restorer line, which corresponds to the C genome in *B. napus* [17]. Moreover, apart from the identified *Rfo* gene [15], other alleles were identified by Wang et al [18,19] and was 1.6 cM distant from *Rfo* locus in Ogu-CMS radish. Further investigations of these alleles revealed a heterozygous type (RsRf3-1/RsRf3-2) encoding PPR proteins were responsible for the fertility of male-sterile radish, and had higher expression and RNA polymerase II occupancy compared with their homozygous alleles (RsRf3-1/RsRF3-1 or RsRf3-2/RsRf3-2) [20]. However, those new identified alleles were studied only in radish. Therefore, associations between these alleles and the *Rfo* gene and their independent functions when transferring into *B. napus* remain unclear.

The introgression of the *Rfo* gene from radish into rapeseed also introduced chromosomes or large amount of radish genome fragments around the *Rfo* locus and possibly elsewhere in the genome. The redundant introgressions are more likely to lead to deleterious genetic characteristics such as poor agronomic performance, increased glucosinolate content, and distorted segregation. The type of cross and mapping population can influence the incidence of genomic regions exhibiting distorted segregation, not to mention the exogenous introgressed or inserted material. However, not many studies linked this aspect to the radish-derived Ogu CMS in *B. napus*, the genetic behaviors of Ogu CMS restorer line as pollen donor or accepter remains unclear and is worthy of investigation. Moreover, molecular marker assisted breeding has been applied throughout the breeding history of Ogu-CMS restorer line in *Brassica* family [11, 21–23]. *Rfo* specific markers in canola were developed by Hu et al [21], while Tian et al [22] and Yu et al. [23] applied to *B. juncea* and *B. oleracea* respectively, enabling previse selection for low glucosinolate within larger population.

Previously we reported a newly-developed Ogu-CMS restorer line (CLR650) from crossing between *Raphanobrassica* (AACCRR, 2n = 56) and *B. napus* using grafting to overcome the incompatible obstacle between these two parents [24]. Further analysis of the updated version of material (CLR6430) developed using the restorer CLR650 through backcross with double-low line and self-pollination was conducted in this study to investigate the chromosomal behaviors and fertility ability. Resequencing was also applied to CLR6430 to explore the detailed information during the introgression from radish fragment into *B. napus*, enabling more precise designs for marker-assisted breeding for the new Ogu-CMS restorer lines application.

## Materials and methods

### Plant material

Using *Raphanobrassica* (AACCRR, 2n = 56) as the donor of Ogu CMS restorer resource, we successfully obtained the hybrids of *Brassica* with *Raphanobrassica* through grafting to overcome the incompatible obstacle between these two parents. After 15 years with high intensive selection, a stable homozygous restorer line CLR650 of Ogu CMS in *B. napus* was bred which can restore the male fertility of progenies from self-pollination and testcross [24]. A new version of Ogu CMS restore lines CLR6430 derived from CLR650 through 2 generations of backcross with double low line 20B and then 4 generations of self-pollination.

### Segregating populations preparation and fertility identification

Two Ogu CMS sterility lines (A-line) 20QA and SC3QA (*rfrf*) were taken as females and crossed with CLR6430 respectively and selfing for F2 and F3 to generate two segregating populations. These two lines were also utilized for back-cross ($BC_{1-2}$) populations and selected for low glucosinolate content concurrently. Reciprocal crosses were conducted between CLR6430 and a general selfing line 20B (The maintainer line of 20QA) and $SC_3$ (The maintainer line of $SC_3QA$).

The fertility identification of all segregating populations was performed around 11 AM on sunny days during flowering stage of plants three times using 1% acetocarmine under a microscope (DS-Ril, Nikon, Japan). And a $F_2$ population of 192 individuals derived from the cross of 20QA×CLR6430 was used for DNA marker testing.

### Chromosome detection

Newly-germinated seed roots and ovaries from young flower buds were used to determine the chromosome numbers of CLR6430. The first 2–3 mm root tips and ovaries were treated with 2 mM 8-hydroxyquinoline for 3–4 h at room temperature and fixed in ethanol: acetic acid (3:1, v/v) before stored at -20˚C. Well-separated chromosome slides were used for chromosome counting (DS-Ril, Nikon, Japan) and then stored for further genomic and fluorescence in situ hybridization (GISH) and fluorescence in situ hybridization of bacterial artificial chromosome (BAC-FISH).

### GISH and BAC-FISH analysis

Plant DNA was extracted from the newly-grown young leaves using DNeasy Plant mini Kit (Qiagen, Hilden, Germany). The BAC clone of restorer gene used *CopyControl™ pCC1BAC™ (Hind III Cloning-Ready) Vector*, and the concentration of chloramphenicol resistance was 12.5μg/mL, with average 95kb inserted in the constructed library. The C genome of *Brassica napus* and BAC clone was labeled by random priming with biotin-11-dUTP (Sabc, China) and

digoxigenin-11-dUTP (Roche, Basel, Switzerland). The A genome of *B. napus* was sheared by boiling for 15 min and used as a block.

The photographs of well-separated chromosome slides were taken under the fluorescence microscopy following the methods describing by Ge and Li [25].

## Collection of re-sequencing sample, DNA extraction and sequencing

Seeds from CLR6430 line (46608, double confirmed with restorer gene) were sown in pots and grow to seedling stages (with 3–4 leaves) before fresh leaf samples were taken for whole genome sequencing (Shanghai OE Biotech. Ltd, China).

The DNA was extracted using DNeasy plant mini kit (Qiagen, Hilden, Germany). The quality and concentration of DNA were tested before digested into fragments.

The library was prepared using TruSeq DNA LT Sample Prep kit (Illumina, SanDiego, CA, USA) with pair-end method.

## Sequencing assembly and mapping

Fragmented DNA sample (300–500 bp) with acquired concentration was sequenced using Illumina HiSeq platform. The Illumina reads generated in this study are available at the website (http://www.ncbi.nlm.nih.gov/sra/SRP159151) with the accession number 46608_4.

After separation of the raw sequence data according to the library-specific barcodes, read qualities of the sequences were initially checked using qualimap (http://qualimap.bioinfo.cipf. es/), and preprocessed by removing low-quality reads (Q20 < 70%), non-ATCG reads and short sequence reads (< 70 bp). After pre-processing, the reads were aligned against the *B. napus* reference genome v4.1 allowing default mismatches [26].

A BLAST search was performed against the *Rfo* gene sequences (GeneBank Accession AJ550021) and revealed that the *Rfo* gene had high sequence identity (over 90%) with certain regions of the linkage group $R_9$ in *R. sativus*, leading further analysis conducted combined with both *B. napus* and $R_9$ from *R. sativus*. Clean reads were then aligned into self-generated reference genome which combine whole genome of *B. napus* and $R_9$ from radish using software BWA-MEM [27]. The coverage and depth of the reads aligned into $R_9$ from radish were analyzed for the homologous regions that introgressed from $R_9$ into *B. napus*.

## Development of specific markers for CLR6430

Based on previous BLAST search results the scaffolds identified from aligned sequences in $R_9$ from radish in reference genome with average coverage over 115 and homologous to $R_9$ from radish were used for developing specific markers for CLR6430. Eighty specific simple sequence repeat (SSR) primer pairs were designed based on the *Raphanus* fragment introgresssed using online marker design website:http://200.137.197.254/~wellington/websat/. The presence of the Ogu CMS and fertility restoration—were verified in every cross generation by PCR. Genomic DNA was extracted from young leaves of field-grown plants, using the cetyltrimethylammonium bromide (CTAB) methods [28].

Each PCR was performed in 96-well PCR plates, with each reaction containing 10-20ng/μl of genomic DNA, 1X standard PCR buffer, 2.5mM MgCl$_2$, 0.1mM dNTP, 1.0 unit Taq polymerase, 0.125μM of forward and reverse primers. PCR amplification used the following cycling parameters: 1 cycle of 5 min at 95˚C; 10 cycles of 30s at 94˚C,45s at 60˚C (next cycle is reduced by 0.5˚C), 30s at 72˚C, then followed by 30 cycles of 30s at 94˚C, 45s at 55˚C, 30s at 72˚C and ending with 10 min at 72˚C). All PCR products were analyzed by electrophoresis in 3.0% agarose gel in 1×TAE buffer and were visualized by ethidium bromide on a digital gel documentation system.

**Table 1. Fertility and sterility results of the offspring crossing with CLR6430 in selfing segregating populations.**

| Population | Generation | $F_1$ Female | Total | Fertility | Sterility | $\chi^2$(P) for 1:2 |
|---|---|---|---|---|---|---|
| A (Rfrf) selfing | F2 | 20QA-1 | 102 | 38 | 64 | 0.71 |
| | | SC$_3$QA-1 | 139 | 49 | 90 | 0.23 |
| | F3 | 20QA-1 | 172 | 61 | 111 | 0.35 |
| | | 20QA-2 | 202 | 72 | 130 | 0.49 |
| | | SC$_3$QA-1 | 158 | 51 | 107 | 0.08 |
| | | SC$_3$QA-2 | 175 | 53 | 122 | 0.73 |

## Analysis of markers in the radish introgression

The specific markers were used to compare CLR6430 with the known rapeseed restorer R2000 [7]. And a radish cultivar "Baiyuchun" was used as the radish comparison. Parts of PCR amplification products were sequenced by Sanger sequencing, and the consensus sequences amplified from the CLR6430 and R2000 were aligned using DNAman ver3.0 (Lynnon BioSoft, Quebec, Canada).

# Results

## Fertility segregation identification

Ogu CMS A-lines 20QA and SC$_3$QA were crossed with the restorer line CLR6430 (20QA×CLR6430 and SC$_3$QA×CLR6430) and then selfed for $F_2$ and $F_3$. Fertility and sterility ratio of tested plants in both F2 and F3 population were closely to 1:2 (Table 1), segregated from the expected 3:1. Similarly, reciprocal crosses were conducted between CLR6430 and 20B and SC$_3$, and fertility and sterility ratio in the backcross population also segregated from expectation, matched closely to 1:4 (Table 2) which distorted from the expected 1:1 ratio. Distorted segregation happened in each population possibly due to the introgressed fragment from radish, causing abnormal segregation of translocated chromosome of *B. napus* during meiosis. Meanwhile, there was no significant difference between the populations from reciprocal crosses [A(*rfrf*) × A(*Rfrf*)] and [A(*Rfrf*) × B(*rfrf*)], indicating no difference of the restorer gene (*Rfo*) passing through male or female donor. However, *Rfo*-specific markers matched precisely to their fertility status in both segregating population ($F_2$ and $F_3$) and back-crossing population (BC$_1$-n), indicating the fertility results from segregating populations were reliable.

## Chromosome number and radish genome introgression

Chromosome numbers were determined in more than 30 individuals of CLR6430. Majority of somatic cells (>82%) in roots had 2n = 38, ranging around 38–40 (Fig 1). The radish

**Table 2. Fertility and sterility results of the offspring crossing with CLR6430 in backcrossing segregating populations.**

| | Crossing | Recurrent parent | Total | Fertility | Sterility | $\chi^2$ (P) for 1:4 |
|---|---|---|---|---|---|---|
| A(rfrf) test crossing | A(rfrf)×A(Rfrf) | 20QA-1 | 158 | 31 | 127 | 0.01 |
| | | 20QA-2 | 207 | 41 | 166 | 0 |
| | | SC$_3$QA-1 | 179 | 36 | 143 | 0 |
| | | SC$_3$QA-2 | 120 | 25 | 95 | 0.05 |
| | A(Rfrf)×B(rfrf) | 20B-1 | 158 | 28 | 130 | 0.51 |
| | | 20B-2 | 243 | 52 | 191 | 0.3 |
| | | SC$_3$-1 | 189 | 37 | 152 | 0.02 |
| | | SC$_3$-2 | 195 | 42 | 153 | 0.29 |

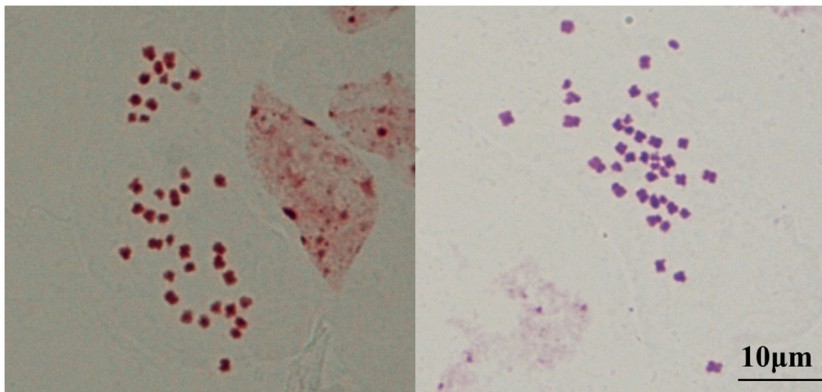

**Fig 1. Chromosome determination of CLR6430 under 40×microscope.** More than 82% of detected chromosome numbers were 2n = 38 of 30 plants from two generation of CLR6430. Scale bar = 10 μm.

introgression was detected using GISH with radish genomic probe and *B. napus* genome as block (Fig 2a and 2b). Significant and stable red fluorescent signals were detected near centromere and satellite regions of several chromosomes, while no red signals were covered as a whole chromosome, showing CLR6430 was introgressed with radish genome without additional radish chromosome.

Restorer gene was detected by FISH using C genome and BAC as fish probe while A genome of *B. napus* was as block (Fig 2c). Green signals were detected consistently in A genome not C genome of *B. napus*, which is different from previous restorer lines (7, 21, 23).

## Re-sequencing of CLR6430 and its genomic assembly

In total, we obtained 134.338 Gb of pair-end sequencing data with a high-quality base ratio of 98.2%, and 71.6% of the reads aligned to the reference with an average coverage of more than 50X (Table 3). While de novo assembling using SOAPdenovo2, all reads were assembled into 568,120 scaffolds and 641,453 contigs (sequence reads >500 bp were summarized).

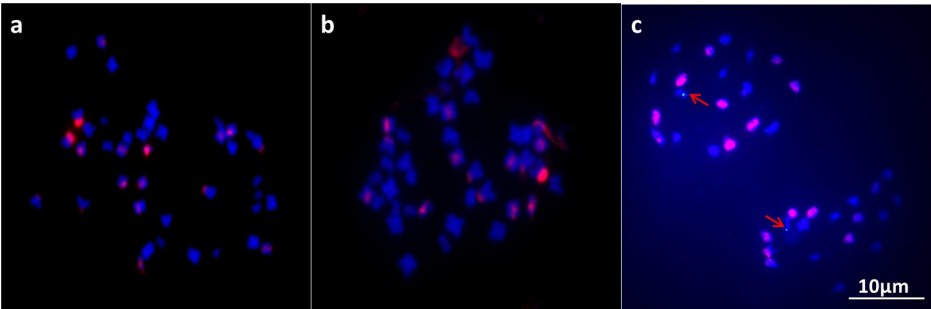

**Fig 2. Fluorescent *in situ* hybridization (FISH) analysis of radish-derived Ogu CMS restorer of *B. napus* (CLR6430).** DAPI (blue) and merged signals (red signals from *R.sativus* probe) from somatic cells are shown in (a) and (b). (a) CLR6430 using 10×*B. napus* genome as block; (b) CLR6430 using 20× *B. napus* genome as block; (c) DAPI (blue) and merged (red signals from C genome of *B. napus* and green signals from restorer gene directed by arrows) from anther of CLR6430 is shown. Scale bar = 10 μm.

**Table 3. Basic sequencing results of CLR6430.**

|  | R1-fastq | R2-fastq |
|---|---|---|
| Total reads number | 447,797,399 | 447,797,399 |
| Clean Reads | 430,273,428 | 430,273,428 |
| Clean reads ratio | 96.09% | 96.09% |
| Total base number | 67169609850(67.169G) | 67169609850(67.169G) |
| Base number in clean reads | 64541014200(64.541G) | 64541014200(64.541G) |
| High quality base number in clean reads | 64031420594(64.031G) | 62812421023(62.812G) |
| High quality base ratio in clean reads | 99.21% | 97.32% |

## Specific molecular marker development

The re-sequencing mapping and alignment results revealed homologous reads derived from $R_9$ in radish assembled into $R_9$ of "reference genome" and the average coverage of the region between 66.13 (Rsa1.0_00994.1) and 138.53cM (Rsa1.0_00176.1) was over 115, much higher than the other regions in $R_9$ group, indicating a high possibility of introgressed fragment from radish into CLR6430. The sequences of this region were then used for developing specific molecular markers in the $F_2$ segregating population (Fig 3). Considering the conservation and variability of SSR markers between the two species, sequences from introgressed fragment were used to develop SSR markers for marker-assisted breeding. Eighty specific SSR markers were developed initially and thirty-two were tested as positive in fertile plants while negative in sterile plants. They were further verified using $F_2$ segregating population and obtained consistent results in correspondence with their fertility conditions in the field. Detailed information of specific SSR markers (Table 4) and sequence of amplified PCR products from Table 4 were presented in Additional Table 1.

## Comparison of the introgression between CLR6430 and R2000 with markers

The above-mentioned 32 markers were used to characterize the CLR6430 in comparison with R2000. The minimum size of the introgressed fragment in CLR6430 was estimated at 72.14 CM (between Rsa1.0_00994.1 and Rsa1.0_00071.1). Comparison to CLR6430, R2000 had five continuously lost markers (CLR9-20, CLR9-21, CLR9-26, CLR9-27, CLR9-30,) (Table 5), estimating the introgressed region about 48.48 CM (between Rsa1.0_00994.1 and

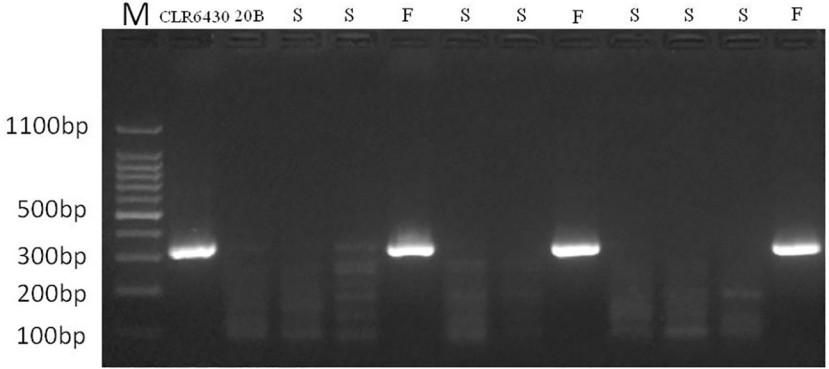

**Fig 3. PCR products amplified by specific SSR markers CLR9-16 in $F_2$ segregating population.** "S" and "F" refer to the samples identified with sterility or fertility during flowering time.

**Table 4. Specific primer pairs designed from the introgressed region from radish in CLR6430.**

| Marker Name | Forward(5'→3') | Reverse(5'→3') | Size of Product (bp) |
|---|---|---|---|
| CLR9-1 | GAACTTATGGCACTCCGATCTC | CGAAGCAAGTAAGAAACACACG | 323 |
| CLR9-2 | GGTCATTTTCTTCCTTGATAGC | ATATAACTAGGTGTTTTGCCCG | 354 |
| CLR9-3 | ATATCGCACGGGTTCCTTAC | CATCCATCAGTTCAATCGGTTA | 439 |
| CLR9-4 | TGTTAAAACCGAGGGAAAAGAG | TCTGCACTTGGGTCACTACAAT | 364 |
| CLR9-5 | CGTAGGGCAGCTTTGATTTTAG | GCTTGTACGCTTCTTCCAGATT | 367 |
| CLR9-6 | AAAGAAGTCTCGCCTGAACAAG | ATGAGAATGGCTAGTCCGGTTA | 349 |
| CLR9-7 | AAGAGAAAACCAGAGCGACAAG | GCAGCGATAGGAAATTGGATAA | 364 |
| CLR9-8 | GCAGCGATAGGAAATTGGATAA | AAGAGAAAACCAGAGCGACAAG | 374 |
| CLR9-9 | CGAACAGAATTGAAACCGAAC | GTTGTACGTCTTCCACTTTCCC | 392 |
| CLR9-10 | CAGAAGCAAGTCGAGAGAGACA | AGGAACCGACATTCAGAGAGAG | 348 |
| CLR9-11 | TATAAAACCTGGGGATTGTTGC | AATTAACCTTGTCGGGTGAAGA | 369 |
| CLR9-12 | AAATGCCTTCCTTGATAACTGG | CGAAGATTTCATTGCTGATACG | 360 |
| CLR9-13 | TGGTGGTGTCTCAAAATGGTA | CATGGTACTCCTGAGCTTATTTG | 385 |
| CLR9-14 | TGCTTTGTATTCATCTCTCCCC | CGACTCTTCAATGTGCATCTCT | 367 |
| CLR9-15 | TGCTAGGGTTCCTCTGGATCT | TCTCCTTCAAAGCAATCTCTCC | 366 |
| CLR9-16 | CTGAGAGGATCATGTTTTGTGC | GCAGAGACTTCTTCACCGTCTT | 311 |
| CLR9-17 | CTTTATCTGCTTCTGCTGTTGC | TTTCTCCCTGATGACCTTTTGT | 346 |
| CLR9-18 | AATAGCTTCCTCACCTGTCACC | GGTTTAGACGGCACCTAGTCAG | 346 |
| CLR9-19 | TGCATACAAACCGAGAATCA | CGGTCTAACATATTGCACATTC | 343 |
| CLR9-20 | GGACAAACAAGGATGGAGTTTC | CCAAATCTGAATGCGAGAGAAT | 535 |

Rsa1.0_00045.1). And the size of fragments amplified by primers pairs CLR9-1, CLR9-6, CLR9-7, CLR9-10 and CLR9-31from CLR6430 and R2000 were different. Besides, the amplified PCR sequences by the new developed SSR markers in this study revealed significant differences between CLR6430 and R2000, presenting various sequence insertion, deletion and replacements (Fig 4).

## Discussion

The radish-derived male sterility has the advantage of complete and stable sterility in *B. napus* and is favored by breeders in commercial production [29]. However, creation of an Ogu-CMS restorer line in *B. napus* remains difficult as no restorer genes were directly found in this species, nor in any Asian radish varieties. Previous research outlined the difficulties and effort in the way of producing an Ogu-CMS *B. napus* restorer line [11], either through cell fusion or consistent backcrossing [30]. In a previous study, Chen *et al* [24] used a *Raphnobrassica* (AACCRR) as donor and successfully obtained the hybrid with *B. napus* through grafting to overcome the incompatibility between these two parents. After 15 years of intensive selection, a homozygous (*RfRf*) restorer line CLR650 was finally obtained. Similar to previous early-version Ogu-CMS restorer line, CLR650 was associated with deleterious agronomic traits, such as high level of glucosinolate content and low seed production. Updated version CLR6430 was generated through multiple generations of back-crossing and selfing, and further corresponding analysis of fertility models, chromosome behaviors, and genomic re-sequencing-based specific markers development were characterized in this study to provide basic information for utilizing of Ogu-CMS restorer lines.

Generally, introgressed fragment from radish is the key part for understanding of the newly created Ogu-CMS in *B. napus*. In previous published lines, at least 50 cM of radish genome was integrated into restored rapeseed based on AFLP marker information [31]. They identified

**Table 5. Specific marker comparison for CLR6430, R2000 and radish in this study.** "+" and "-" were used to describe the results as positive or negative. "Baiyuchun" was the name of a radish cultivar.

| Radish genome | Region (cM) | Marker Name | Scaffold Number | CLR6430 | R2000 | Baiyuchun | Note |
|---|---|---|---|---|---|---|---|
| R9 | 66.13 | CLR9-32 | | - | - | + | |
| R9 | 66.97 | CLR9-31 | | + | + | + | |
| R9 | 72.64 | CLR9-13 | | + | + | + | |
| R9 | 74.73 | CLR9-11 | Scaffold510 | + | + | + | |
| R9 | 76.06 | CLR9-33 | | + | + | + | |
| R9 | 76.06 | CLR9-34 | | + | + | + | |
| R9 | 84.38 | CLR9-10 | | + | + | + | |
| R9 | 93.38 | CLR9-14 | Scaffold4378 | + | + | + | *Rfo* region |
| R9 | 93.38 | CLR9-1 | Scaffold662 Scaffold33209 Scaffold42248 Scaffold61106 Scaffold176555 Scaffold251659 | + | + | + | *Rfo* region |
| R9 | 93.38 | CLR9-2 | | + | + | + | *Rfo* region |
| R9 | 93.38 | CLR9-3 | Scaffold80976 | + | + | + | *Rfo* region |
| R9 | 93.38 | CLR9-4 | Scaffold66484 | + | + | + | *Rfo* region |
| R9 | 93.38 | CLR9-5 | Scaffold6789 Scaffold208888 | + | + | + | *Rfo* region |
| R9 | 93.38 | CLR9-22 | | + | + | + | *Rfo* region |
| R9 | 93.38 | CLR9-23 | Scaffold80 | + | + | + | *Rfo* region |
| R9 | 93.38 | CLR9-6 | Scaffold80 Scaffold100822 | + | + | + | *Rfo* region |
| R9 | 93.38 | CLR9-7 | Scaffold80 Scaffold214757 | + | + | + | *Rfo* region |
| R9 | 93.38 | CLR9-8 | Scaffold80 Scaffold214757 | + | + | + | *Rfo* region |
| R9 | 93.38 | CLR9-9 | Scaffold80 | + | + | + | *Rfo* region |
| R9 | 94.92 | CLR9-24 | | + | + | + | |
| R9 | 94.92 | CLR9-25 | | + | + | + | |
| R9 | 103.77 | CLR9-15 | Scaffold10090 Scaffold111532 | + | + | + | |
| R9 | 115.45 | CLR9-16 | Scaffold7186 Scaffold53692 | + | + | + | |
| R9 | 115.45 | CLR9-17 | Scaffold62 | + | + | + | |
| R9 | 115.45 | CLR9-18 | Scaffold4699 | + | + | + | |
| R9 | 115.45 | CLR9-19 | Scaffold62 | + | - | + | |
| R9 | 118.34 | CLR9-20 | | + | - | + | |
| R9 | 119.83 | CLR9-21 | | + | - | + | |
| R9 | 120.43 | CLR9-26 | Scaffold13452 | + | - | + | |
| R9 | 121.78 | CLR9-27 | | + | - | + | |
| R9 | 130.79 | CLR9-28 | | + | + | + | |
| R9 | 132.17 | CLR9-29 | Scaffold83707 | + | + | + | |
| R9 | 138.23 | CLR9-30 | | + | - | + | |

**Fig 4. Sequence comparison between CLR6430 and R2000 in the region near *Rfo* gene, which was amplified by CLR9-7 marker.**

the region of *Arabidopsis* genome syntenic to the *Rfo* locus, then carried out fine mapping of the *Rfo* gene in a segregating radish population to find the likely ortholog of the *Rfo* gene. In recent years, along with the development of second-generation sequencing, radish genome has also been available [32–34] and useful for the development of specific markers for breeding purposes. In our study, CLR6430 was re-sequenced and aligned against a combined reference genome, which was the whole *B. napus* genome plus an additional chromosome from radish

that contained the *Rfo* gene. The high coverage sequences assembled into R9 in the reference genome indicating the most likely homologous regions from radish introgressd into *B. napus*, which is the region between 66.13 cM (Rsa1.0_00994.1) and 138.53cM (Rsa1.0_00176.1). Therefore, there is about 72.4 cM of radish genome integrated into CLR6430. And the following design of SSR markers for CLR6430 was also based on the re-sequencing and alignment results, providing solid and more precise information for later improvement and utilization of CLR6430.

Secondly, creation of an ideal restorer line for Ogu-CMS *B. napus* remains difficult. BAC-FISH results (Fig 2) has demonstrated restorer gene from Radish located in A genome rather than N19 from C genome as previously reported (7, 21, 23), therefore, the introgression of radish genome into *Brassica napus* is different. After obtaining homozygous (*RfRf*) instead of heterozygous (*Rfrf*) in the loci of the restorer gene, we still have to break the linkages with some deleterious traits such as high glucosinolate content and low seed production. R2000, the widely accepted Ogu-CMS restorer line developed by INRA, has been utilized for three-line system production of hybrid rapeseed.

Meanwhile, the comparison between CLR6430 and R2000 in this study has shown significant differences, revealing the different origin. CLR6430 has improved performances in terms of glucosinolate content and seed production ability compared to its earlier version CLR650, while the introgressed fragment from radish remain long and require further improvement. In this case, developed markers for R2000 or other reported Ogu-CMS restorer were not applicable for CLR6430, hence self-designed markers based on resequencing results are meaningful for further selection and improvement for CLR6430.

Besides, segregation distortion was found in CLR6430 in selfing and backcrossing populations. Distorted segregation is the deviation of the observed genotypic ratios from the expected frequencies based on Mendelian's laws of inheritance, which is considered as an evolutionary force and associated with genetic factors involved in reproduction and fitness [35]. Several mechanisms of segregation distortion have been reported in plants, such as chromosomal rearrangement and genomic interactions causing zygotic abortion, hybrid sterility, haploid induction and restriction of gene introgression [36]. In our case, distorted segregation happened stably in each population possibly due to the introgressed fragment from radish, causing abnormal segregation of translocated chromosome of *B. napus* during meiosis, which could affect the estimation of map distances and the order of markers when many distorted markers are presented. Other reason for this phenomenon could be the selection of both CLR650 and CLR6430 were based on high-pressure selection of *Rfo* gene and the introgressed fragment along it remains large.

The radish-derived male sterility has the advantage of complete and stable sterility in *B. napus* and was favored by breeders in commercial production [37]. As one of the most important rapeseed market, China has rarely utilized this Ogu-CMS system in hybrid seed production, mainly due to the lack of self-developed restorer line and patent restrictions applied from other restorer sources such as R2000 and SRF developed by INRA and Pioneer Hibred, respectively. Making good use of CLR6430 could be the breakthrough for future application of Ogu-CMS system in this market.

In summary, the DNA sequence surrounding the *Rfo* in CLR6430 is different from that of R2000, and resequencing-based *Rfo*-related markers presented a more efficient method for the improvement of CLR6430. However, the glucosinolate content remains higher than market level, and marker-assisted improvement still at early stage for seed settings and other agronomic traits. Moreover, the genetic mechanisms underlying the observed segregation distortion for *Rfo* in the inbreeding and backcross offspring requires further investigation. Further

studies are required to locate and reduce the size of the introgression in CLR6430 for better breeding outcomes.

## Supporting information

**S1 Table. Sequence of amplified PCR products from self-designed SSR markers from Table 4.**
(DOCX)

## Acknowledgments

We thank Prof Tingdong Fu for his generosity of providing the initial bridging material and his valuable advice on the progress. We also thank Ms. Qian Yang for her assistance in data analysis of the resequencing data of CLR6430.

## Author Contributions

**Conceptualization:** Tonghua Wang, Weijiang Chen.

**Data curation:** Tonghua Wang, Yiming Guo, Zengxiang Wu, Shengqian Xia.

**Formal analysis:** Tonghua Wang.

**Funding acquisition:** Mei Li.

**Investigation:** Yiming Guo, Shengqian Xia, Weijiang Chen.

**Methodology:** Zengxiang Wu.

**Resources:** Shuijin Hua, Weijiang Chen.

**Software:** Shengqian Xia.

**Supervision:** Jinxing Tu, Mei Li, Weijiang Chen.

**Validation:** Zengxiang Wu.

**Visualization:** Tonghua Wang.

**Writing – original draft:** Tonghua Wang, Yiming Guo.

**Writing – review & editing:** Yiming Guo, Jinxing Tu, Mei Li.

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
