## [Decision Letter · Decision Letter 0]

27 Apr 2020

PONE-D-20-03673

Genetic characterization of a new radish introgression line carrying the restorer gene for Ogura CMS in Brassica napus

PLOS ONE

Dear Dr. Guo,

Thank you for submitting your manuscript to PLOS ONE. After careful consideration, we feel that it has merit but does not fully meet PLOS ONE’s publication criteria as it currently stands. Therefore, we invite you to submit a revised version of the manuscript that addresses the points raised during the review process.

We would appreciate receiving your revised manuscript by Jun 11 2020 11:59PM. To enhance the reproducibility of your results, we recommend that if applicable you deposit your laboratory protocols in protocols.io, where a protocol can be assigned its own identifier (DOI) such that it can be cited independently in the future. For instructions see: http://journals.plos.org/plosone/s/submission-guidelines#loc-laboratory-protocols

We look forward to receiving your revised manuscript.

Kind regards,

Yong Pyo Lim

Academic Editor

PLOS ONE

Reviewers' comments:

Reviewer's Responses to Questions

**Comments to the Author**

1. Is the manuscript technically sound, and do the data support the conclusions?

Reviewer #1: Yes

Reviewer #2: Yes

2. Has the statistical analysis been performed appropriately and rigorously? 

Reviewer #1: N/A

Reviewer #2: Yes

3. Have the authors made all data underlying the findings in their manuscript fully available?

Reviewer #1: Yes

Reviewer #2: Yes

4. Is the manuscript presented in an intelligible fashion and written in standard English?

Reviewer #1: Yes

Reviewer #2: Yes

5. Review Comments to the Author

Reviewer #1: In their study, Guo et al. present an investigation into the Rfo introgression in CLR6430. Unlike other restorer line, they find Rfo on N19 (C9?) in a differently sized introgression.

I can't find any fault with the general outline of the study or the findings, only minor things (below).

It seems that the short reads generated in this study are openly available, however, the genome assembly generated is not. Will the authors deposit this assembly somewhere?

Some minor things:

Line 29, typo, 'introgresssed'

Line 170 - the link in the PDF is broken and points to 'http://wsmartins.net/websat/ with default', should be without the 'with default'

Line 218 - incomplete sentence? 'Restorer gene was detected by FISH using C genome and BAC fish probe while A genome of B. napus as block'

Line 233, Table 3 - typo, 'Cleaned eads'

Line 265, why is Baiyuchun mentioned only once here? Why is it not in the methods?

Line 287, typo - 'Raphnobrassica'

There are a few more smaller typos which should have been caught by rigorous spell checking.

Reviewer #2: This manuscript 'Genetic characterization of a new radish introgression line carrying the restorer gene

for Ogura CMS in Brassica napus' is based on well defined problem and conducted in a profound way. It is a significant improvement from the earlier available studies. This should be accepted for publication after incorporating the following corrections:

i. Introduction: Explain the importance of restores in oilseed brassicas

ii. In the materials and method, the population developed using the CMS lines and restores and maintainers with restorers were mentioned. However, it has not been made clear how these population were used for this study. The method used in Table 5 is not mentioned clearly in this section.

iii. Results: Explained appropriately, however, it is difficult to understand how the different populations were used in validation of the designed markers and segregation ratio. Explain these portion elaborately.

iv. Discussion: This section need improvement in terms of the works done and its significance. This part should explain all the experiments conducted supported by sufficient number of citations.

Comments in the manuscript is also attached for further refernce.

6. PLOS authors have the option to publish the peer review history of their article (what does this mean?). If published, this will include your full peer review and any attached files.

Reviewer #1: No

Reviewer #2: Yes: Dr. Shyam Sundar Dey

---

## [Author Response · Author response to Decision Letter 0]

7 Jun 2020

Reviewer #1: In their study, Guo et al. present an investigation into the Rfo introgression in CLR6430. Unlike other restorer line, they find Rfo on N19 (C9?) in a differently sized introgression.

I can't find any fault with the general outline of the study or the findings, only minor things (below).

It seems that the short reads generated in this study are openly available; however, the genome assembly generated is not. Will the authors deposit this assembly somewhere?

Thank you for these positive comments and careful reminder. We have deposited the genome assembly to the NCBI database under the same BioProject number PRJNA488570.

Some minor things:

Line 29, typo, 'introgresssed'

Revised to “introgressed”

Line 170 - the link in the PDF is broken and points to 'http://wsmartins.net/websat/ with default', should be without the 'with default'

The website has been updated and now changed to ‘http://200.137.197.254/~wellington/websat/’, 

Line 218 - incomplete sentence? 'Restorer gene was detected by FISH using C genome and BAC fish probe while A genome of B. napus as block'

Revised to ‘Restorer gene was detected by FISH using C genome and BAC clone as probes while A genome of B. napus as block’

Line 233, Table 3 - typo, 'Cleaned eads'

Revised to ‘clean reads’

Line 265, why is Baiyuchun mentioned only once here? Why is it not in the methods?

The relative information has added into the methods part.

Line 287, typo - 'Raphnobrassica'

Revised to ‘Raphanobrassica’.

There are a few more smaller typos which should have been caught by rigorous spell checking.

We have gone through the manuscript thoroughly to minimize the errors in grammar and spelling.

Reviewer #2: This manuscript 'Genetic characterization of a new radish introgression line carrying the restorer gene

for Ogura CMS in Brassica napus' is based on well defined problem and conducted in a profound way. It is a significant improvement from the earlier available studies. This should be accepted for publication after incorporating the following corrections:

i. Introduction: Explain the importance of restores in oilseed brassicas

One sentence explaining the importance of restores in oilseed Brassicas has been added into the Introduction part.

ii. In the materials and method, the population developed using the CMS lines and restores and maintainers with restorers were mentioned. However, it has not been made clear how these population were used for this study. The method used in Table 5 is not mentioned clearly in this section.

Thank you for the reminder, the information of the radish cultivar “Baiyunchun” has been added in the methods part for Table 5.

iii. Results: Explained appropriately, however, it is difficult to understand how the different populations were used in validation of the designed markers and segregation ratio. Explain these portion elaborately.

The process of validation of designed markers in tested and verified in the F2 segregating population was further explained in the context.

iv. Discussion: This section need improvement in terms of the works done and its significance. This part should explain all the experiments conducted supported by sufficient number of citations.

Thank you for the suggestion. We revised and rearranged several parts in this part while added a few more citations.

Comments in the manuscript is also attached for further refernce.

---

## [Decision Letter · Decision Letter 1]

6 Jul 2020

Genetic characterization of a new radish introgression line carrying the restorer gene for Ogura CMS in Brassica napus

PONE-D-20-03673R1

Dear Dr. Guo,

We’re pleased to inform you that your manuscript has been judged scientifically suitable for publication and will be formally accepted for publication once it meets all outstanding technical requirements.

Kind regards,

Yong Pyo Lim

Academic Editor

PLOS ONE

Additional Editor Comments (optional):

Reviewers' comments:

Reviewer's Responses to Questions

**Comments to the Author**

1. If the authors have adequately addressed your comments raised in a previous round of review and you feel that this manuscript is now acceptable for publication, you may indicate that here to bypass the “Comments to the Author” section, enter your conflict of interest statement in the “Confidential to Editor” section, and submit your "Accept" recommendation.

Reviewer #1: All comments have been addressed

Reviewer #2: All comments have been addressed

2. Is the manuscript technically sound, and do the data support the conclusions?

Reviewer #1: Yes

Reviewer #2: Yes

3. Has the statistical analysis been performed appropriately and rigorously? 

Reviewer #1: N/A

Reviewer #2: Yes

4. Have the authors made all data underlying the findings in their manuscript fully available?

Reviewer #1: Yes

Reviewer #2: Yes

5. Is the manuscript presented in an intelligible fashion and written in standard English?

Reviewer #1: Yes

Reviewer #2: Yes

6. Review Comments to the Author

Reviewer #1: I'm happy with the way my recommendations were implemented. The data has been added to SRA and typos have been fixed. Thanks!

Reviewer #2: The manuscript entitled, ' Genetic characterization of a new radish introgression line carrying the restorer gene for Ogura CMS in Brassica napus , has been improved and revised as per the suggestions of the reviewers. Therefore, in the present format it can be accepted for publication.

7. PLOS authors have the option to publish the peer review history of their article (what does this mean?). If published, this will include your full peer review and any attached files.

Reviewer #1: No

Reviewer #2: **Yes: **Dr. Shyam Sundar Dey

---

## [Editor Report · Acceptance letter]

9 Jul 2020

PONE-D-20-03673R1 

Genetic characterization of a new radish introgression line carrying the restorer gene for Ogura CMS in Brassica napus 

Dear Dr. Guo:

I'm pleased to inform you that your manuscript has been deemed suitable for publication in PLOS ONE. Congratulations! Your manuscript is now with our production department. 

Kind regards, 

on behalf of

Dr. Yong Pyo Lim 

Academic Editor

PLOS ONE